# Effects of Early Vocal Contact in the Neonatal Intensive Care Unit: Study Protocol for a Multi-Centre, Randomised Clinical Trial

**DOI:** 10.3390/ijerph18083915

**Published:** 2021-04-08

**Authors:** Manuela Filippa, Elisa Della Casa, Roberto D’amico, Odoardo Picciolini, Clara Lunardi, Alessandra Sansavini, Fabrizio Ferrari

**Affiliations:** 1Department of Psychology and Educational Sciences, University of Geneva, 24, rue General Dufour, 1211 Geneva, Switzerland; 2Department of Social Sciences, University of Valle d’Aosta, Str. Cappuccini, 2, 11100 Aosta, Italy; 3Neonatal Intensive Care Unit, Department of Medical and Surgical Science of the Mother, Children and Adults, University of Modena and Reggio Emilia, Via del Pozzo 41, 41124 Modena, Italy; dellacasa.elisa@policlinico.mo.it (E.D.C.); fabrizio.ferrari@unimore.it (F.F.); 4Department of Diagnostics, Clinical and Public Health Medicine, University of Modena and Reggio Emilia, Via del Pozzo 41, 41124 Modena, Italy; rdamico@unimore.it; 5Pediatric Physical Medicine & Rehabilitation Unit, IRCCS Ca’ Granda Ospedale Maggiore Policlinico, Via Francesco Sforza, 35, 20122 Milan, Italy; odoardo.picciolini@policlinico.mi.it; 6Department of Neurosciences, Psychology, Drug Research and Children’s Health, University of Florence, Viale Pieraccini, 6, 50139 Florence, Italy; clara.lunardi@unifi.it; 7Department of Psychology “Renzo Canestrari”, University of Bologna, Viale Berti Pichat 5, 40127 Bologna, Italy; alessandra.sansavini@unibo.it

**Keywords:** preterm infants, early vocal contact, heart rate variability, neurobehavioral development

## Abstract

Preterm infants are at risk for developing altered trajectories of cognitive, social, and linguistic competences compared to a term population. This is mainly due to medical and environmental factors, as they are exposed to an atypical auditory environment and simultaneously, long periods of early separation from their parents. The short-term effects of early vocal contact (EVC) on an infant’s early stability have been investigated. However, there is limited evidence of its impact on the infant’s autonomic nervous system maturation, as indexed by heart rate variability, and its long-term impact on infant neurodevelopment. Our multi-centric study aims to investigate the effects of EVC on a preterm infant’s physiology, neurobehaviour, and development. Eighty stable preterm infants, born at 25–32 weeks and 6 days gestational age, without specific abnormalities, will be enrolled and randomised to either an intervention or control group. The intervention group will receive EVC, where mothers will talk and sing to their infants for 10 min three times per week for 2 weeks. Mothers in the control group will be encouraged to spend the same amount of time next to the incubator and observe the infant’s behaviour through a standard cluster of indicators. Infants will be assessed at baseline; the end of the intervention; term equivalent age; and 3, 6, 12, and 24 months corrected age, with a battery of physiological, neurobehavioral, and developmental measures. Early interventions in the neonatal intensive care unit have demonstrated effects on the neurodevelopment of preterm infants, thereby lowering the negative long-term effects of an atypical auditory and interactional environment. Our proposed study will provide new insight into mother–infant early contact as a protective intervention against the sequelae of prematurity during this sensitive period of development. Early intervention, such as EVC, is intuitive and easy to implement in the daily care of preterm infants. However, its long-term effects on infant neurodevelopment and maternal sensitivity and stress are still unclear. Trial Registration: NCT04759573, retrospectively registered, 17 February 2021.

## 1. Background

Preterm birth rates are continuously increasing in most countries and over 500,000 infants are born prematurely every year [1]. Despite rapid advances in technology, the number of preterm infants with low gestational age (GA) at birth, who demonstrate short- and long-term sequelae of prematurity before reaching school age remains high [2]. Approximately 40% of low birth weight preterm infants experience a complex spectrum of unfavorable neurodevelopment outcomes [3] compared with their full-term peers. Thus, prematurity is of great concern for health policies, in both low and high-income countries [4,5]. The aforementioned impaired development of preterm infants is associated with medical factors and partly a consequence of their atypical early-life environment, including uncontrolled sensory inputs, pain stressors, and early maternal separation [6,7,8] during their hospitalisation in the neonatal intensive care unit (NICU). 

Developmental care practices aim to minimise the stress of the NICU environment through specific interventions, which may include control of external sensory stimuli [9] pain and stress management, and increased family involvement in the preterm infant’s care [10]. Health policies for guidelines in the NICU are still inhomogeneous, especially regarding parental presence in the NICU and family-centred care. Specific guidelines have recently been published with the support of an interdisciplinary group of experts, comprising paediatricians, parents, neonatologists, psychologists, and therapists [11]. These, hopefully, will be broadly adopted in NICUs in the forthcoming years.

### 1.1. Medical End Environmental Factors Lead the Preterm Infants’ Development

In addition to critical medical conditions, there is evidence that the NICU environment, to which preterm infants are exposed during the critical periods of brain maturation, can interfere with development. Early stressful experiences, particularly those related to an altered sensory exposure [12], painful procedures [7], and separation from primary caregivers [13] are considered among the most deleterious environmental factors affecting preterm infant’s development. The experience of this atypical environment during the sensitive and vulnerable period of brain growth may have negative cascading effects on their neuropsychological development [14].

The first year of life coincides with maximal brain growth, but is also the most vulnerable period in development. Preterm birth exposes infants to a severe mismatch with foetal brain development, which presents a significant challenge for a developing brain. The third trimester of gestation is a period of striking sequential brain changes [15]. Neurogenetic events occur across the cerebral compartments of the brain, including white matter tract growth, where the outgrowth of axons involved in pathway finding, guidance, and resting periods present a sensitive adaptation that can be affected by changes in the environment [16]. For early preterm infants, an important feature is the vulnerability of the thalamocortical pathways within the crossroads and sagittal strata. It is likely that this frenetic development of the brain and the diffuse subtle brain abnormalities found in preterm infants are influenced by environmental inputs [17]. 

In these phases, brain growth is activity-dependent, and the underlying mechanisms are regulated by early cortical synaptic activity. In animal models, researchers investigating the mechanisms of environmental enrichment or deprivation have reported a powerful effect of experience on brain development [18,19]. Therefore, the integrity of the brain processes depends upon the presence of the correct neural elements appearing at the appropriate developmental moments [20], in a delicate equilibrium between environmental stimulation and endogenous brain activity.

### 1.2. Atypical Auditory Environment in the NICU and Atypical Voice Treatment for Preterm Infants

The maturation of neuronal circuits during the third trimester of gestation is highly dependent on the intrinsic and extrinsic multisensory activity. The ability of foetuses and newborns to perceive the different sound levels in terms of energy (i.e., dBA) of the auditory stimulus [21], and sound pitches develops over time in utero and after full-term birth, due to the types and frequencies of sensory stimuli. For example, the tonotopic organisation of the cochlea continues maturing until early infancy and in cases of early exposure to loud background noise (e.g., preterm birth), it may fail to tune the hair cells of the cochlea, leading to frequency discrimination [22].

Instead of their mother’s womb, preterm infants spend the third trimester of gestation in a bright and unpredictable noisy place, the NICU. Stress during their stay in the NICU has been associated with more brain structural and functional alterations [23]. Conversely, protective actions aiming to avoid deleterious sensory stimuli and support a sensory-enriched environment during this period have demonstrated enhanced functional and structural brain development [24,25].

The auditory environment in the NICU is not always appropriate for sustaining the preterm infant’s auditory development; loud and unpredictable sounds from monitors, warnings associated with medical emergencies, hospital architectural characteristics (i.e., windows of the incubators, cabinet doors), and continuous background sounds (i.e., ventilators, CPAP) are common elements in everyday hospital soundscapes, often exceeding the recommended levels of sound pressure [24]. As such, acoustic criteria for the NICU have been proposed by Philbin and Evans [26] and clear recommendations for noise control in the NICU have been formulated [27].

The auditory development of preterm infants can encounter several problems in coordinating with the NICU’s soundscape. This is because the perception of a noise signal is not adult-like until late childhood and an immature auditory system can only perceive stimuli, even if salient (e.g., maternal voice) with an adequate signal-to-noise ratio [28]. Moreover, the predictability and organisation of background sounds affect attention, especially in infants and children, who require more repetitive and redundant auditory information than adults [29]. 

In the delicate balance between the inadequate soundscape of the NICU and a plastic and sensitive developing brain, whether and how the auditory experience of preterm infants in the NICU is controlled is crucial in early interventions.

Preterm infants at term-equivalent age treat voices atypically [30]. However, this discussion and interpretation are still debated. Bartha-Doering and colleagues [30] reported that full-term newborns demonstrate a significantly greater hemodynamic response to forward speech compared with backward speech within the left hemisphere, which is typically involved in language treatment. This suggests an altered development of the functional network underlying language acquisition in preterm infants, which is what further delays preterm infants from developing voice sensitivities in language areas. According to the authors this alteration in developing voice sensitivities in language areas is one of the potential reasons for further language delays in the preterm infants’ population. Adam-Darque and colleagues [31] interpreted their findings on additional cortical regions involved in voice processing and a late mismatch response to maternal voice in preterm versus term infants as potential positive signs for an earlier development of voice discrimination abilities in preterm infants, whose capacity to process voices may be enhanced due to an earlier airborne exposure to voices.

The amount of auditory experience adequate for perception and language development is unknown [32]. The debate surrounding early auditory enrichment with recorded stimulations is still open [33] and requires further investigation on the specific and distinctive tasks in preterm versus term voice perception and discrimination. This is for basic aspects such as forward and backward voices and early pitch (frequency), prosody (dynamic changes of pitch in time), melody, speech rhythm detection, and voice discrimination abilities.

### 1.3. Early Protective Interventions

A number of experimental studies have found that relation-based early interventions can impact infant development, mainly if they provide high-quality and ecologically pervasive actions [34], with the main effects lasting for the first two years of life [35]. Changes in sensory inputs and mother-driven environmental enrichment at this stage can affect brain development and stress regulation. Preterm birth carries severe risk factors for a dysfunctional mother-infant interaction cycle with the potential for developing lower maternal sensitivity and responsiveness and higher maternal control [36]. Both these forms of dysfunctional regulatory patterns show a lack of synchronicity, which is an essential element in the interactive regulation between developing attachment behaviour [37]. In the regulatory process of human attachment, maternal vocalisations to infants are “biologically relevant signals” [38] and affect brain development. The mother’s external regulation of the infant’s state during particular critical periods may represent the essential factor influencing the experience-dependent growth of brain areas, eliciting high levels of metabolic energy for the tuning of the brain cortical-subcortical circuits involved in processing socioemotional information [39].

Several studies investigated the effects of the live maternal voice—namely singing mothers supported by professionals—during music therapy interventions in the NICU, primarily suggesting it as an early music intervention in the frame of environmental enrichment and infant emotional development [40]. In these studies, the maternal singing, supported by instruments and by the music therapist, has been primary used to promote parent-infant interaction [41,42], to sustain preterm infants’ regulation and vital signs [43,44], and as a strategy to involve families in the early care in the NICU [45,46,47]. 

At this stage, early maternal care can affect an infant’s ability to react and positively adapt to stressful conditions: skin-to-skin contact [48], early vocal contact [49], maternal tactile stimulation [50], and active parental closeness.

### 1.4. EVC in the NICU Impacts Both the Auditory and Relational Environment

EVC actively involves parents in emotional and meaningful vocal contact with their preterm infants during hospitalisation in the NICU [49]. It sustains the preterm infant’s physiological stability, with a significant decrease in critical events (e.g., bradycardia, apnoea, and hypoxia) and increase in a calm awake state [51]. When preterm infants in the NICU are exposed to more adult voices, they show higher seven- and 18-month corrected age (CA) Bayley-III language and cognitive scores [52]. Low parental presence and the consequent decreased interaction and lower language exposure contribute to sensory deprivation experienced by infants in neonatal units, which impacts their brain structure and neurodevelopmental outcomes [12]. 

Filippa and colleagues illustrated the effects of live maternal vocalisations, singing and speaking, on the preterm infant’s behaviour [53] and reciprocally, the association between specific infant pro-social behaviour, such as smiling and opening eyes, and acoustic characteristics of the preterm infant-directed speech and singing [54]. The father’s vocal contact also impacts the preterm behavioural organisation and state in a manner similar to the mother’s voice, with calm awakening effects [55]. Changes in either parent’s vocal qualities are related to the preterm infant’s behaviour [56], indicating a reciprocal responsivity during early face-to-face interaction. 

The advances in intensive care have reduced neonatal mortality and severe outcomes such as cerebral palsy and sensory deficiencies. Conversely, there has been no concomitant reduction in consequential problems such as neurological disorders, subnormal cognitive function, language disorders, behavioural problems, learning disabilities, psychopathology (e.g., autism), and physical disability. The long-term developmental impairments of preterm babies are associated with perinatal, social, and relational risk factors. Further studies are needed to investigate the benefits of early healthcare and family-centred interventions for reducing the long-term impact of neonatal hospitalisation and minimising the negative impact due to invasive health interventions and stressful procedures. The lack of validated family-centred early interventions and the invasiveness of health technology assessment methods in preterm infants constitute the principal needs and challenges for further research. Moreover, no previous studies have evaluated the effect of EVC on a preterm infant’s autonomic maturation and at neurobehavioral and developmental levels.

### 1.5. Aims 

The proposed study aims to test the effects of EVC on infants born prematurely at 25–32 weeks and 6 days GA. The effects on newborns will be assessed at physiological (primary outcome), neurobehavioral, and developmental levels during the intervention, at term equivalent age, and at 6, 12, and 24 months CA. The effect of EVC on maternal stress at hospital discharge and on the mother’s presence in the NICU will also be assessed.

## 2. Hypothesis

We hypothesise that EVC can enrich the preterm infants’ sensory and interactional experience and that this mother-driven environmental enrichment impacts the infant’s physiology by enhancing cardiorespiratory maturation and autonomic regulation that persist during interventions and possibly heart rate variability (HRV). 

HRV is representative of the functional state of an infant’s autonomic nervous system (ANS). ANS maturation is accompanied by increasing HRV with a concomitant augmentation of parasympathetic activity. A preterm infant’s GA correlates with HRV; preterm infants demonstrate lower HRV parameters than term infants [57]. Early interventions aimed at involving parents in engaging with preterm infants have shown important effects on an infant’s HRV [58] and respiratory sinus arrhythmia [59], the two indices of autonomic regulation and maturation of the ANS. 

We also hypothesise that an interactive and relational-based enriched auditory environment can impact an infant’s neurobehavioral development, as evidenced by the General Movements score (for details, see Secondary Outcome Measures).

Specific interventions directed at improving early care in the NICU, such as the Newborn Developmental Care and Assessment Program [60], skin-to-skin care [61], and massage therapy [61,62], have shown significant effects on neurodevelopmental outcomes. Research suggests that early interventions aimed at reducing the sequelae of prematurity can have positive effects, especially in the short-term periods [63,64]. However, many interventions lack long-term assessments, particularly on the infants’ neural and behavioural development [65].

Thirdly, in a long-term prospective design, we will test if earlier and regular exposure to salient linguistic stimuli in interactive contexts will impact preterm infant development, particularly the cognitive and linguistic performance, at 6, 12, and 24 months CA. Adult language input is important for healthy language development and preterm infants are at risk for long-lasting language delays [66,67]. Thus, we expect to find a direct effect of the intervention on the aforementioned developmental domains.

Finally, we expect that EVC will reduce maternal stress in the NICU and increase maternal presence during hospitalisation by involving mothers in early interactions with their infants. Over the course of the past 50 years, several studies have confirmed that early mother-infant separation leads to adverse physiological and behavioural consequences that persist throughout life [68,69].

In general, we expect results that support the importance of EVC in the NICU and suggest that EVC may provide an effective, feasible, and easy-to-implement approach to improve outcomes in preterm infants.

## 3. Methods

### 3.1. Design

A four-site randomised controlled trial will be conducted to investigate the short- and long-term physiological and neurobehavioral effects of EVC.

### 3.2. Study Setting

Infants will be recruited from four university hospitals in Italy: Careggi University Hospital (Florence), NICU Fondazione IRCCS Ca’ Granda Ospedale Maggiore Policlinico (Milan), Modena University Hospital (Modena), and Bologna University Sant’Orsola-Malpighi Hospital (Bologna), and randomised to either an intervention or control group. Infants in the intervention group will receive EVC from the mother, whereas in the control group, mothers will be encouraged to spend the same amount of time next to the incubator, observing their infant’s behaviour through a standard cluster of indicators. The randomisation sequence, unique for the four centres, is led by the Statistical Office of the University of Modena and Reggio Emilia.

### 3.3. Participants

Eighty preterm infants, born at 25–32 weeks and 6 days GA, will be recruited from the four centres (20 per centre). Recruitment will be undertaken in each centre by a trained research assistant (RA), who will review birth records daily. After obtaining permission from the attending physicians of both the mother and infant, the RA will invite qualifying families to participate and will obtain written informed consent from those enrolling. Infants will be then assigned to the EVC intervention or control group using a randomised design, stratifying infants by gender and GA (see Randomisation). Mothers whose infants are assigned to the control group will be asked to spend the same amount of time as that of mothers in the intervention group, observing their infants’ spontaneous behaviour, with the subsequent compilation of an observation grid developed ad hoc, according to few indicators drawn from The Neonatal Behavioural Assessment Scale. The mothers whose infants are assigned to the control group will be asked not to vocally interact with them during the observational intervention.

Recording sessions will take place midway between the feeding cycles, more than 1 h after the feed and more than 4 h after the last medical exam.

### 3.4. Patient and Public Involvement Statement

Patients were not involved in the design and conduct of the study, nor in the choice of outcome measures. They will be involved at the first stage of the recruitment (see *Participants*).

### 3.5. Inclusion Criteria

The inclusion criteria for the newborns will be: GA between 25 + 0 and 32 + 6 weeks at birth; Apgar score ≥ 7 at 10 min; birth weight between 3rd and 97th percentiles; birth cranial circumference greater than 10th percentile; periventricular leukomalacia (PVL) grade 1; intraventricular haemorrhage (IVH) grade 1–2; and hypoglycaemia, hyponatraemia, and hypocalcaemia are permitted, provided they are not persistent and severe. 

### 3.6. Exclusion Criteria

The exclusion criteria for the newborns will be: PVL grade III and IV, IVH grade III and IV, sepsis (vertical and horizontal), congenital malformations and/or genetic abnormalities, requiring respiratory support with high flow/nCPAP repeated apnoea associated with bradycardia and fall of saturation, hyaline membrane disease, respiratory distress syndrome, and hyperbilirubinaemia requiring exchange transfusions during hospitalisation. 

Lack of informed consent will exclude the infant from the study.

The exclusion criteria for the mothers will be presence of depressive symptoms, drug abuse, and age below 18 years.

### 3.7. Sample Size and Power Analysis

The main outcome measures for the primary outcome will be the low-frequency (LF) and high-frequency (HF) power of the HRV measure. We expect the baseline measurement mean value (±standard deviation) of LF and HF power for preterm infants to be 31 ± 4.9 Hz and 15 ± 2.6 Hz, respectively [58]. We assume that the treatment in the experimental population is effective if the mean values of LF and HF power reach 25 ± 4.9 and 17 ± 2.4 Hz, respectively. 

With an alpha setting at 0.025 and beta setting at 0.80, it is necessary to enroll 29 patients per group. Assuming a 20% loss, we aim to enroll at least 80 patients total (40 per group).

### 3.8. Randomisation

A centralised patient randomisation will be performed using a secure, web-based randomisation system, through which the study investigator of each centre can register new patients and obtain the treatment arm assignment.

A block allocation sequence with variable block sizes (block size 2–4), stratified by GA and gender, will be generated using Stata statistical software (StataCorp, College Station, TX, USA) with the “ralloc” command. In case of twins, randomisation will be conducted at the level of the family (i.e., twins will be randomised to the same group). The allocation sequence will be concealed from the investigators.

### 3.9. Intervention

This intervention will start soon after birth, once the inclusion criteria are met, and will continue over 2 weeks during hospitalisation, minimising the effect of mother–infant separation in the NICU. In any case, participants will be excluded if they reach the inclusion criteria at more than 36 weeks of GA at test, thus when it’s presumed that they will not complete the two weeks of intervention. 

The EVC will take place in the hospital room while infants are in their individual incubators or open cribs. In the intervention group, mothers will be asked to speak and sing to their infants continuously over a 10 min period for each type of intervention (20 min in total). Mothers will be asked to talk in their native language and to sing familiar songs, while observing their infant’s reactions. The order of the two vocalisations, speaking and singing, will be reversed in the next intervention.

EVC will be performed by mothers three times per week for 2 weeks, more than 1 h after afternoon feeding. It will begin when the newborns are in an active sleep state, in calm awake state or in active awake state, but not in deep sleep or crying. 

The intervention will not begin during deep sleep state, in order to protect the infant sleep cycles and his stability [27]. The calm awake state is the most suitable for beginning early sensory interventions and interaction with the family, in stable preterm infants [70]. However, there is the need not to overwhelm the infant in this precious attentional state, where the brain expects sensory experience to grow and mature [20].

Preterm infants will be enrolled from 25 + 0 to 32 + 6 weeks of GA, following the established inclusion criteria.

Two fixed cameras will record the facial and upper torso movements of the mothers and infants. Mothers will be free to choose the content and style of their speech and singing, but will be asked to refrain from touching the infant, while maintaining eye contact with them. They will be told to place their face in front of the incubator opening so as to be as close as possible to the infant’s head. 

To improve adherence to the intervention protocol, concomitant medical procedures will not be permitted during the intervention. 

### 3.10. Control Group

Mothers in the control group will be encouraged to spend the same amount of time next to the incubator, observing the infant’s behaviours through a standard cluster of indicators. It’s important that mothers in the control group are sustained in the presence and in the observation tasks, in order to minimize the negative effects of separation in the NICU.

## 4. Measures

### 4.1. Baseline Measures

#### 4.1.1. Infant’s Measures

As a physiological assessment, electrocardiogram (ECG) recordings will be collected during a 20-min baseline period, in the absence of the mothers, at the same hour as that of the intervention. We will also use a qualitative assessment of general movements (GMs) to assess behaviour. A video of the baby’s general movements will be collected in the baseline period. The video coding will be performed on blinded videos by two independent researchers.

#### 4.1.2. Maternal Measures

Baseline demographic data will be collected from the digital recorded data for each patient. In addition, a socio-economic status questionnaire will be administered to each participant and the Socio Economical Score (SES) will be calculated. To determine baseline stress, the Parental Stressor Scale (PSS-NICU) questionnaire will be administered to the mothers before intervention (T0) and before discharge, both in the intervention and control groups. The PSS-NICU [71,72] aims at assessing the parental perception of stressors derived from the physical and psycho-social environment of the NICU across three domains: their parental role, their infant’s behaviour and appearance, and the sights and sounds in the NICU. For each domain, a mean score will be assessed, and a final composite stress score will be calculated from the mean values of the single scores. Each mother, at each time point, will receive a final score for the single questionnaire (range 0–10). 

A questionnaire on musical experience [73] will be administered at T0. The questionnaire gives three scores for musical expertise, musical habits in everyday life, and music as an emotional experience. 

#### 4.1.3. Environmental Measures

In each centre, the background noise levels will be acquired via a calibrated sound level meter (Voltcraft Phonometer SL-10; Conrad Electronic, Hirschau, Germany) in the room where the intervention will take place and inside the incubator, 10 cm from the newborns’ head. This measure will be assessed at every intervention session to ensure that the mother’s voice is audible to the newborn (i.e., it exceeds the background noise of 10 dBA). No further analyses will be performed on these values.

## 5. Outcome Measures

### 5.1. Primary Outcome Measure

#### Physiological Level

During the interventions, time and frequency analysis of HRV will be applied to the following parameters: LFpower (power in the LF band), HFpower (power in the HF band), LFnu (normalised power in the LF band [LF power/(LFpower + HFpower)]); HFnu (normalised power in the HF band [HF power/(LFpower + HFpower)]); LF% (percentage power in the LF band [LF power/total power]), HF% (percentage power in the HF band [HF power/total power]), and LF/HF (ratio between LF power and HF power). The change from baseline will be calculated across the three conditions (i.e., singing, speaking, and control).

HRV will be measured through the acquisition of the ECG signal, via electrodes placed on the infant’s chest by routine medical devices. Subsequently, the ECG signal will automatically be recorded by the IxTrend Software (Ixellence GmbH, Wildau, Germany). Moreover, the heart rate (beats per minute), oxygen saturation (%), and all critical events (hypoxemia, oxygen saturation <80%; bradycardia, heart rate <80 bpm) will be acquired during the different conditions using the relevant medical apparatus. 

The physiological level is strictly dependent on the infant’s behavioural state (i.e., quiet, active, indeterminate, awake, or crying), which will be assessed offline from the video-recorded sessions by blinded coders, according to the Prechtl scale [8]. Measures of physiological parameters will be collected over the intervention period, as well as 20 min before and after the mother’s intervention.

### 5.2. Secondary Outcome Measures

Secondary outcome measures are presented in Figure 1.

#### 5.2.1. Infant’s Measures

Neurobehaviourally, GMs [8] will be measured at T0 and T1 (at term equivalent age) in both groups. Infants will be reassessed at 3 months. The GMs quality from video recording will be scored according to the Ferrari optimality score [74]. Two blinded coders will attribute a single final score for each infant at each time point. Inter-rater reliability will be assessed, and the mean scores will be compared between the intervention and control groups at each time point.

Developmental Level: Developmentally, the Griffiths Mental Development Scales (GMDS) [75] and a routine neurological examination (EON) will be administered to infants by trained researchers at 6 (T2), 12 (T3), and 24 (T4) months CA. 

These GMDS assess infant development across five main areas (locomotor, personal and social skills, hearing and language, eye and hand coordination, and performance), providing a general developmental quotient (DQ) of infant’s abilities and five sub-scale quotients (SQ).

The EON will be assessed with a final positive or negative score. In case of a negative score, the infant will be excluded from further analyses. The Italian version of the MacArthur–Bates Communicative Development Inventory [76,77] will be completed by parents at 12 (Gestures and Words Form) and 24 (Words and Sentences Form) months CA. Furthermore, the expressive and receptive language test, Parole in Gioco [78], will be administered to infants by trained researchers at 24 months. Each child, at each time point, will receive a final score for each questionnaire, measured as a discrete numeric value. 

Next, an audiological evaluation using an auditory brain system response will be performed within 3 months of CA and repeated at 12 and 24 months with behavioural audiometry techniques. A positive or negative evaluation will be obtained. In the case of negative audiological evaluation, the infant will be excluded from analyses.

#### 5.2.2. Maternal Measures

##### Stress Measures

To determine stress, questionnaires administered at baseline will also be administered at hospital discharge. The final PSS-NICU scores will be compared between the intervention and control groups, and between two different time points (i.e., pre- and post-intervention).

##### Maternal Presence in the NICU

Measures of attendance and the frequency and/or time spent using maternal self-report forms will be filled out after each visit to the NICU. The mothers in both groups will be asked to enter information about the frequency and duration of their visits to the NICU. All questionnaires on maternal presence in the NICU will be collected at discharge. The final data will be the duration of maternal presence in the NICU (min) for each day of hospitalisation and the frequency of visits. 

## 6. Blinding

The nature of the present trial does not allow blinding for the included patients for either group. Physiological data analysts will be blinded after their assignment to interventions. Double blinded coding will be assured for video analysis; two trained blinded coders will perform the GMs analysis through silent videos. Cognitive and linguistic tests will be administered and coded blindly. 

## 7. Potential Confounds

A limitation of the study is the potential contamination between the intervention and control groups. Whenever possible, the mothers of the two groups will not simultaneously be present in the same hospital room.

## 8. Statistical Analyses

### 8.1. Primary Outcome

The modulation of vagal activity will be tracked as a function of time during the interaction between the infants and mothers, by calculating the LF and HF power of the HRV and investigating respiratory sinus arrhythmia. The acquired autonomic signals (e.g., ECG, PPG, and breath signal) will be analysed to evaluate the differences between the physiological responses related to the different experimental groups. Standard analysis techniques that involve the extraction of parameters in the time and frequency domain will be applied in the first instance. 

Intra-group analysis (considering the control group and the intervention group separately) will be performed with the Friedman test by comparing the values of HRV parameters extracted from the three sessions (pre, yes and post). Post-hoc correction (Bonferroni method) will be applied. Inter-group analysis (between the control group and the intervention group) will be performed with the Kruskal–Wallis test by comparing HRV parameters of the first five-minute window of each session (pre, yes and post) between the two groups. Finally, innovative methods of analysis aimed at applying the chaos theory to physiological signals will be used to study the complexity of autonomic dynamics and the nonlinear coupling between the autonomic subsystems. These nonlinear analysis techniques will be applied both univariate (using a single signal, preferably HRV) and multivariate (e.g., considering HRV and pulse rate variability at the same time) domains.

### 8.2. Secondary Outcomes

To assess the effect of EVC on the neurobehavioral and developmental outcomes of newborns, as well as on maternal stress and presence in the NICU, the values for the different scales and questionnaires will be compared (at the time points specified in the Secondary Outcome Measures) between the intervention and control groups. Confounding variables will be entered in the analysis (i.e., familiar socio-economical score, gestational age at birth) to partial out any effect which they could have on the difference between the outcome measures. For continuous variables the Wilcoxon-Mann-Whitney test will be applied, while for categorical variables the chi-squared test (or Fisher’s exact test where appropriate) will be applied. 

## 9. Data Management

The project coordinator, Dr. Manuela Filippa, will supervise the data collection and will ensure the data security management according to the actual rules. A data repository will be created with anonymous codes, ensuring the data protection for all patients enrolled. Only the principal investigators of each center and the project coordinator will have access to the personal data of the patients and the data will be collected by the research assistant anonymously. The project coordinator will prevent the publication, modification, deletion and copying of personal health-related data without authorization or inadvertently. Finally, traceability will be ensured in all phases of the study (recruitment, data collection, data storage, data analysis, result dissemination). If adverse events occur, they will be reported to the project coordinator and immediate actions will be implemented.

## 10. Discussion

Preterm infants demonstrate developmental delays in a wide range of domains when compared to their term peers. Most of these gaps can be reduced by well-tailored and timely early interventions in the NICU. There is an urgent need for simple and cost-effective early intervention methods that actively involve parents in the early care of their preterm infants. This study will provide new insights into mother-infant early contact as protective intervention against the sequelae of prematurity during the sensitive period of development. 

In preterm infants, EVC has been found to be beneficial for the mother and infant, by increasing infant’s stabilisation and maternal emotion during vocal contact. Most previous studies have focused on the short-term effects of EVC in preterm infants. However, its long-term effects on infant neurodevelopment and on maternal sensitivity and stress still require exploration. EVC may be a promising early intervention for parents and their preterm infants, consistent with the recommendation of individualised and family-based developmental care. 

## Figures and Tables

**Figure 1 ijerph-18-03915-f001:**
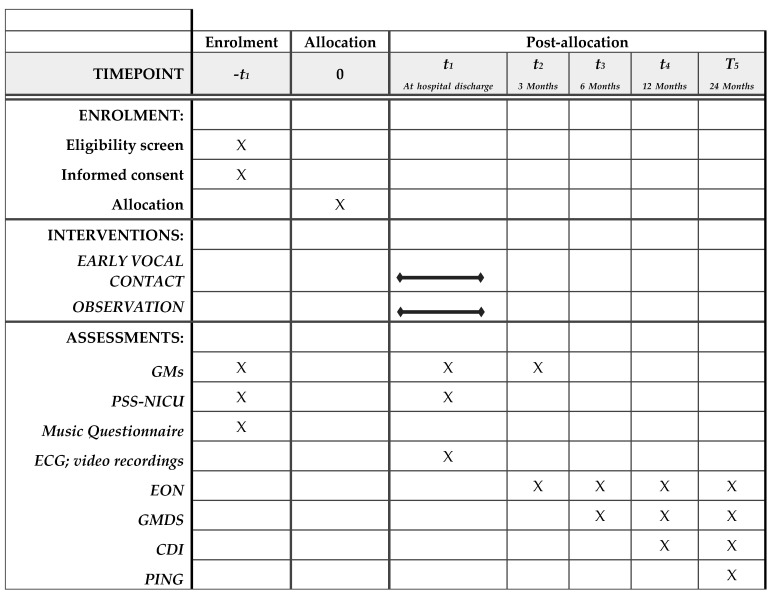
Enrolment, interventions, and assessments.

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
