# Peer review of "Effects of Early Vocal Contact in the Neonatal Intensive Care Unit: Study Protocol for a Multi-Centre, Randomised Clinical Trial"

_ijerph, 2021, doi:10.3390/ijerph18083915_

Round 1
Reviewer 1 Report
This study is well designed and stands to make an important contribution to the growing body of literature on best practice in the NICU. There are only a few points to address that would strengthen the proposed study.
1) There is a large body of literature on voice and neonatal care that could be cited for a more thorough literature review (Shoemark; Loewy; Haslback for example.
2) Authors could more clearly define the behavioral states (323-324) under "intervention", specifically reporting the contraindications of disrupting quiet alert states in which critical neural growth is occurring.
Author Response
Please, see the attachment

Reviewer 2 Report
I would suggest to add a specific statistical analysis section in the method to cover the details of how to analyze different measures, and control covariates.
Besides, for long-term maturation/cognitve development, family socioeconomic factors may also need to be included.
Author Response
Please, see the attachment

Reviewer 3 Report
The authors featured a study protocol that they consider that will provide new insight into mother-infant early contact as a protective intervention against the sequelae of prematurity during this sensitive period of development.
The manuscript is well written, could be interesting for some readers and it is well situated within the Journal Scope. Although I have some minor comments that could improve the final version of the manuscript.
I congratulate the authors for the writing and presentation of the proposed protocol, which I believe can positively affect the development of premature infants.
- Line 77-77: a reference is needed to support these sentences.
- Line 289: some information is missed.
- Line 313: “This intervention will start soon after birth”: authors should be more precise. Although they indicated “once the inclusion criteria are met”, they should indicate a range of days or a period that should not be exceeded to include the participants in the study, thus allowing the recruitment process to be standardized.
Author Response
Please, see the attachment
